

# The HOSPITAL score and LACE index as predictors of 30 day readmission in a retrospective study at a university-affiliated community hospital

Robert Robinson and Tamer Hudali

Department of Internal Medicine, Southern Illinois University School of Medicine, Springfield, IL, United States

Corresponding author
Robert Robinson,
rrobinson@siumed.edu

## ABSTRACT

**Introduction**. Hospital readmissions are common, expensive, and a key target of the Medicare Value Based Purchasing (VBP) program. Validated risk assessment tools such as the HOSPITAL score and LACE index have been developed to identify patients at high risk of hospital readmission so they can be targeted for interventions aimed at reducing the rate of readmission. This study aims to evaluate the utility of HOSPITAL score and LACE index for predicting hospital readmission within 30 days in a moderate-sized university affiliated hospital in the midwestern United States.

**Materials and Methods**. All adult medical patients who underwent one or more ICD-10 defined procedures discharged from the SIU-SOM Hospitalist service from Memorial Medical Center (MMC) from October 15, 2015 to March 16, 2016, were studied retrospectively to determine if the HOSPITAL score and LACE index were a significant predictors of hospital readmission within 30 days.

**Results**. During the study period, 463 discharges were recorded for the hospitalist service. The analysis includes data for the 432 discharges. Patients who died during the hospital stay, were transferred to another hospital, or left against medical advice were excluded. Of these patients, 35 (8%) were readmitted to the same hospital within 30 days. A receiver operating characteristic evaluation of the HOSPITAL score for this patient population shows a C statistic of 0.75 (95% CI [0.67–0.83]), indicating good discrimination for hospital readmission. The Brier score for the HOSPITAL score in this setting was 0.069, indicating good overall performance. The Hosmer–Lemeshow goodness of fit test shows a $\chi^2$ value of 3.71 with a $p$ value of 0.59. A receiver operating characteristic evaluation of the LACE index for this patient population shows a C statistic of 0.58 (95% CI [0.48–0.68]), indicating poor discrimination for hospital readmission. The Brier score for the LACE index in this setting was 0.082, indicating good overall performance. The Hosmer–Lemeshow goodness of fit test shows a $\chi^2$ value of 4.97 with a $p$ value of 0.66.

**Discussion**. This single center retrospective study indicates that the HOSPITAL score has superior discriminatory ability when compared to the LACE index as a predictor of hospital readmission within 30 days at a medium-sized university-affiliated teaching hospital.

**Conclusions**. The internationally validated HOSPITAL score may be superior to the LACE index in moderate-sized community hospitals to identify patients at high risk of hospital readmission within 30 days.

## INTRODUCTION

Hospital readmissions are common and expensive, with nearly 20% of Medicare patients being readmitted to a hospital within 30 days of discharge at an overall cost of nearly 20 billion US dollars per year (*Jencks, Williams & Coleman, 2009*). Because of this high frequency and cost, hospital readmissions within 30 days of discharge are a target for health care cost savings in the Medicare Value Based Purchasing (VBP) program. The VBP aims to incentivize hospitals and health systems to reduce readmissions through reductions in payments to hospitals with higher than expected readmission rates (*Centers for Medicare and Medicaid Services, 2016*). Because of the VBP initiative, health care organizations are investing considerable resources into efforts to reduce hospital readmission.

The underlying risk factors for hospital readmission are diverse. Studies have identified age, race, having a regular health care provider, major surgery, medical comorbidities, length of hospital stay, previous admissions in the last year, failure to transfer important information to the outpatient setting, discharging patients too soon, the number of medications at discharge, and many other risk factors for hospital readmission within 30 days (*Auerbach et al., 2016*; *Picker et al., 2015*; *Hasan et al., 2010*; *Silverstein et al., 2008*). Despite identifying with these risk factors, healthcare providers have poor accuracy in predicting which patients are at high risk of hospital readmission without a risk assessment tool (*Allaudeen et al., 2011*).

Readmission risk assessment can be accomplished with a variety of assessment tools that range from multidisciplinary patient interviews to simple screening tools using a handful of variables (*Zhou et al., 2016*; *Kansagara et al., 2011*; *Silverstein et al., 2008*; *Smith et al., 2000*). These tools use risk factors such as age, ethnicity, socioeconomic status, severity of illness, previous hospitalizations, and other factors to predict who is likely to be readmitted.

The easy to use HOSPITAL score is one such screening tool (*Donzé et al., 2013*). The HOSPITAL score uses 7 readily available clinical predictors to accurately identify patients at high risk of potentially avoidable hospital readmission within 30 days. This score has been internationally validated in a population of over 100,000 patients at large academic medical centers (average size of 975 beds) and has been shown to have superior discriminative ability over some prediction tools (*Kansagara et al., 2011*; *Donzé et al., 2013*; *Donzé et al., 2016*).

Another simple prediction model for predicting hospital readmission which uses both administrative and primary data is the LACE index (*Van Walraven et al., 2010*). The LACE index uses four variables to predict the risk of death or nonelective 30-day readmission after hospital discharge among both medical and surgical patients: length of stay (L), acuity of the admission (A), comorbidity of the patient (C) and emergency department use in the duration of 6 months before admission (E) (*Van Walraven et al., 2010*). This model has been internally validated using data collected from 4,812 patients discharged from 11 community hospitals in Ontario, and it was externally validated using administrative data
collected randomly from 1,000,000 discharges also in Ontario (*Van Walraven et al., 2010*). The LACE index has variable results in the literature outside Ontario. The LACE index has been shown to have moderate discrimination in studies conducted in North America with over 26,000 Medicare admissions (*Garrison et al., 2016*), 110,000 discharges in the Chicago, Illinois area (*Tong et al., 2016*) and 600 patients in a community hospital (*Spiva et al., 2016*). The LACE index had fair discrimination in a study of 5,800 patients in Singapore (*Low et al., 2015*) and poor discrimination in a study done on about 500 patients in UK with an average age of 85 years old (*Cotter et al., 2012*).

A direct comparisons between the HOSPITAL score and LACE index in a nationwide sample of Medicare patients admitted to the hospital for any reason showed no significant differences (*Garrison et al., 2016*). This is contrasted by comparisons of the HOSPITAL score and LACE index from Denmark (*Cooksley et al., 2015*) and Switzerland (*Aubert et al., 2016*) which indicate the HOSPITAL score has superior performance in predicting the risk of hospital readmission. This study aims to conduct a similar comparison of the utility of the HOSPITAL score and LACE index as predictors of hospital readmission within 30 days of discharge in a moderate-sized (507 bed) university-affiliated hospital located in the United States of America.

## MATERIALS AND METHODS

All adult medical patients discharged from the SIU-School of Medicine (SIU-SOM) Hospitalist service from Memorial Medical Center from October 15, 2015 to March 16, 2016, were studied retrospectively to determine if the HOSPITAL score or LACE index were significant predictors of any cause (avoidable and unavoidable) hospital readmission within 30 days. Exclusion criteria were transfer to another acute care hospital, leaving the hospital against medical advice, or death. The any cause readmission within 30 days of hospital discharge endpoint was selected because it is the measure used by the Medicare VBP.

Memorial Medical Center is a 507 bed not-for-profit university-affiliated tertiary care center located in Springfield, Illinois, USA. The SIU-SOM Hospitalist service is the general internal medicine residency teaching service staffed by board certified or board eligible hospitalist faculty. Patients for the hospitalist service are primarily admitted via the hospital emergency department or transferred from other regional hospitals with acute medical issues. Elective hospital admissions are extremely rare for this service.

Data on age, gender, diagnosis related group (DRG), International Classification of Disease (ICD) diagnosis codes, emergency department visits in the last 6 months, length of stay, hospital readmission within 30 days, and the other variables in the HOSPITAL score (Table 1) and LACE index (Table 2) were extracted from the electronic health record in a de-identified manner for analysis. Laboratory tests were infrequently obtained on the day of hospital discharge for hemoglobin (10%) and sodium (53%). Missing laboratory data (hemoglobin and sodium from the day of discharge) were coded to be in the normal range.

The study hospital does not have a distinct oncology admitting service. To address the increased risk of readmission in oncology patients found in other studies using the

**Table 1   HOSPITAL Score.**

| Attribute | Points if positive |
|---|---|
| Low hemoglobin at discharge (<12 g/dL) | 1 |
| Discharge from an Oncology service | 2 |
| Low sodium level at discharge (<135 mEq/L) | 1 |
| Procedure during hospital stay (ICD10 Coded) | 1 |
| Index admission type urgent or emergent | 1 |
| Number of hospital admissions during the previous year | |
| 0–1 | 0 |
| 2–5 | 2 |
| >5 | 5 |
| Length of stay ≥ 5 days | 2 |

**Table 2   LACE index.**

| Attribute | Points if positive |
|---|---|
| Length of stay | |
| Less than 1 day | 0 |
| 1 day | 1 |
| 2 days | 2 |
| 3 days | 3 |
| 4–6 days | 4 |
| 7–13 days | 5 |
| ≥14 days | 7 |
| Acute or emergent admission | 3 |
| Charlson comorbidity index score | |
| 0 | 0 |
| 1 | 1 |
| 2 | 2 |
| 3 | 3 |
| ≥4 | 5 |
| Visits to emergency department in previous 6 months | |
| 0 | 0 |
| 1 | 1 |
| 2 | 2 |
| 3 | 3 |
| ≥4 | 4 |

HOSPITAL score, this study classified patients with oncology related diagnosis ICD codes to have been discharged from an oncology service. This reflects local practice patterns where hospitalists often admit patients to the general medicine service for oncologists. Because data is only available from the study hospital, readmissions at other hospitals will not be detected.

Institutional review board review for this study was obtained from the Springfield Committee for Research Involving Human Subjects. This study was determined not to

meet the criteria for research involving human subjects according to 45 CFR 46.101 and 45 CFR 46.102.

## STATISTICAL ANALYSIS

The HOSPITAL score and LACE index were investigated as predictors of any cause hospital readmission within 30 days. Qualitative variables were compared using Pearson chi$^2$ or Fisher's exact test and reported as frequency (%). Quantitative variables were compared using the non-parametric Mann–Whitney $U$ or Kruskal–Wallis tests and reported as mean ± standard deviation.

The HOSPITAL score and LACE index were calculated for each admission. HOSPITAL scores of 0–4 points were classified as low risk for readmission (5%), 5–6 points intermediate risk (10%), and 7 or more points as high risk (20%) based on the initial validation study of the HOSPITAL score (*Donzé et al., 2013*).

LACE indexes ranged from 0–19, with an expected probability for readmission of 0 to 43.7% based on the initial validation study of the LACE score (*Van Walraven et al., 2010*). We choose the LACE index of 10 or more as the cut point for high risk of admission, the predicted risk of readmission in the initial study mentioned above was 12.2% to 43.7%.

These readmission risk predictions were used to calculate a Brier score.

Most statistical analyses were performed using SPSS version 22 (SPSS Inc., Chicago, IL, USA).

The Brier score was calculated with R version 3.3.1 (R Foundation for Statistical Computing, Vienna, Austria).

Two sided $P$-values < 0.05 were considered significant.

## RESULTS

During the study period (154 days), 463 discharges were recorded for the SIU-SOM Hospitalist service. The analysis includes data for the 432 discharges for 376 individual patients that met inclusion criteria (Fig. 1). Of these discharges, 35 (8%) were readmitted to the same hospital within 30 days. The population that was readmitted within 30 days of discharge includes 29 unique patients. The overall study population was 48% female, had an average age of 62 years, and spent an average of 7.8 days in the hospital.

The patients readmitted as compared to the patients who were not readmitted were younger, more frequently readmitted to the hospital in the last year and had higher HOSPITAL scores. Those differences were statistically different. Other baseline characteristics including the LACE score were not statistically different between the two groups as shown in Table 3. All patients were deemed as urgent or emergent admissions and had an ICD10 coded procedure during hospitalization.

A receiver operating characteristic (ROC) evaluation of the HOSPITAL score for this population showed a C statistic of 0.75 (95% CI [0.67–0.83]) indicating good discrimination for hospital readmission (Fig. 2). The Brier score for the HOSPITAL score in this setting

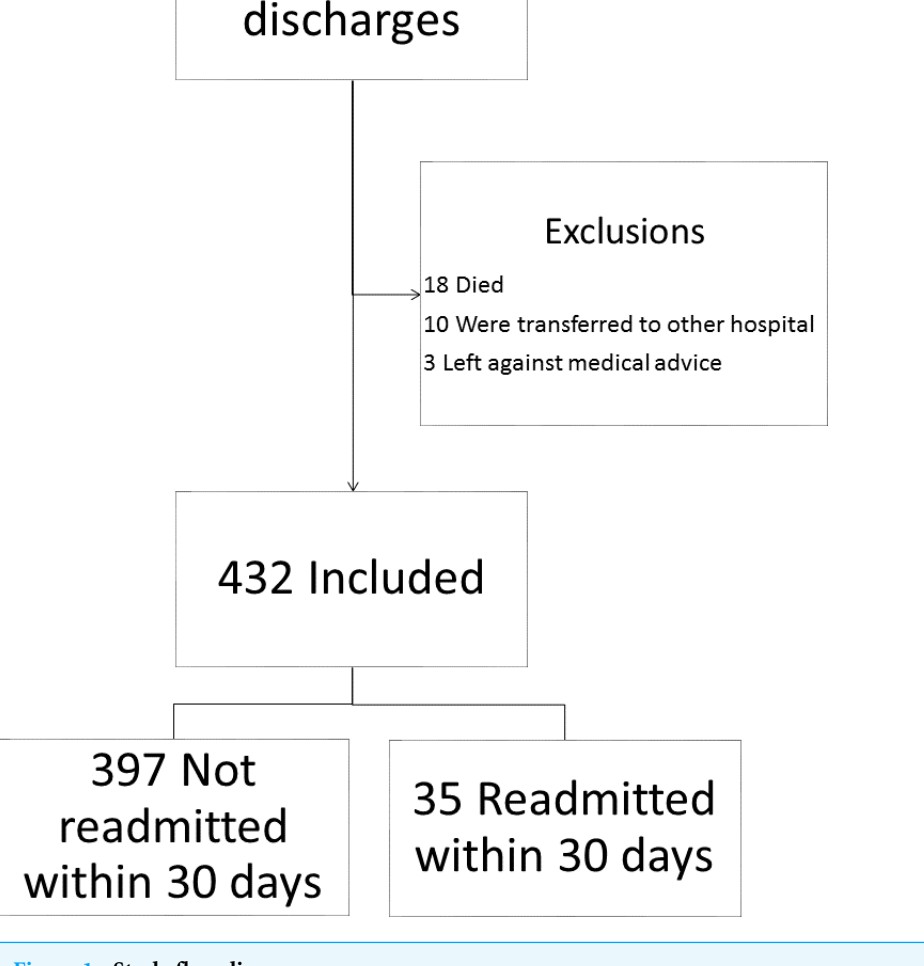

**Figure 1 Study flow diagram.**

was 0.069, indicating good overall performance. The Hosmer–Lemeshow goodness of fit test showed a $\chi^2$ value of 3.71 with a $p$ value of 0.59.

A ROC evaluation of the LACE index for this population showed a C statistic of 0.58 (95% CI [0.48–0.68]) indicating poor discrimination for hospital readmission (Fig. 3). The Brier score for the LACE index in this setting was 0.082, indicating good overall performance. The Hosmer–Lemeshow goodness of fit test showed a $\chi^2$ value of 4.97 with a $p$ value of 0.66.

## DISCUSSION

In this study we aimed at validating both the HOSPITAL score and the LACE index for predicting all cause hospital readmission in our study population.

This single center retrospective study indicates that the HOSPITAL score has good discrimination and calibration to predict all cause hospital readmissions within 30 days

**Table 3** Baseline characteristics of the study population by 30 day readmission status.

| Characteristic | Not readmitted within 30 days $n = 397$ | Readmitted within 30 days $n = 35$ | |
|---|---|---|---|
| Age, mean (SD) | 62 (15.7) | 56 (14.9) | 0.01 |
| Female | 193 (49%) | 15 (43%) | 0.60 |
| Urgent or emergent admission | 397 (100%) | 35 (100%) | |
| Discharge from oncology service | 41 (10%) | 3 (9%) | 1.00 |
| Length of stay ≥ 5 days | 246 (62%) | 23 (66%) | 0.72 |
| Hospital admissions in the last year | 2.3 (3.0) | 5.2 (1.7) | <0.001 |
| Emergency department visits in last 6 months | 2.3 (1.9) | 3.3 (3.6) | 0.031 |
| An ICD10 coded procedure during hospitalization | 397 (100%) | 35 (100%) | |
| An ICD10 coded cancer diagnosis | 32 (8%) | 3 (7%) | 0.742 |
| Low hemoglobin level at discharge (<12 g/dL) | 26 (6%) | 4 (11%) | 0.28 |
| Low sodium level at discharge (<135 mEq/L) | 86 (22%) | 9 (26%) | 0.53 |
| Charlson comorbidity index score (SD) | 4.4 (3.0) | 5 (3.7) | 0.43 |
| HOSPITAL score ≥ 5 (high risk) | 235 (55%) | 31 (86%) | <0.001 |
| LACE index ≥ 10 (righ risk) | 337 (79%) | 32 (89%) | 0.20 |

for a medical hospitalist service at a university-affiliated hospital. On the other hand, the LACE index showed poor discrimination to predict all cause hospital readmission within 30 days for the same study population. The study population contains patients who were admitted more than one time within the study period, six of those individuals were readmitted within 30 days of hospital discharge. Inclusion of these patients is essential for this analysis because it reflects the criteria used by the Medicare Value Based Purchasing program to assess readmission rates (*Centers for Medicare and Medicaid Services, 2016*). The rate of readmission within 30 days in this population was 8%, which is less than the 20% rate of readmission seen in Medicare patients in a nationwide sample (*Jencks, Williams & Coleman, 2009*).

This data for all-causes of hospital readmission is comparable to the discriminatory ability of the HOSPITAL score in the international validation study (C statistics of 0.75 vs. 0.71) conducted at considerably larger hospitals (975 average beds vs 507 at Memorial Medical Center) for potentially avoidable hospital readmissions (*Donzé et al., 2016*) and data from a nationwide Medicare cohort (C statistics of 0.75 vs. 0.675) investigating all cause hospital readmissions (*Garrison et al., 2016*).

The HOSPITAL score had good overall performance in this setting with a Brier score of 0.10 and a Hosmer–Lemeshow goodness of fit test showing a $\chi^2$ value of 1.63 with a *p* value of 0.20. The Brier score from this study is similar to the score reported in the validation study (*Donzé et al., 2016*). The validation study had a superior goodness of fit test, likely reflecting the considerably larger sample size (*Donzé et al., 2016*).

The data shown in our study indicated that the LACE index had a poor discriminatory ability of predicting all-cause 30 hospital readmissions (C statistics of 0.58, 95% CI [0.48–0.68]). This differs from the original validation study of the LACE index (C statistics of 0.58 vs. 0.684) which showed moderate discrimination for early death or readmission

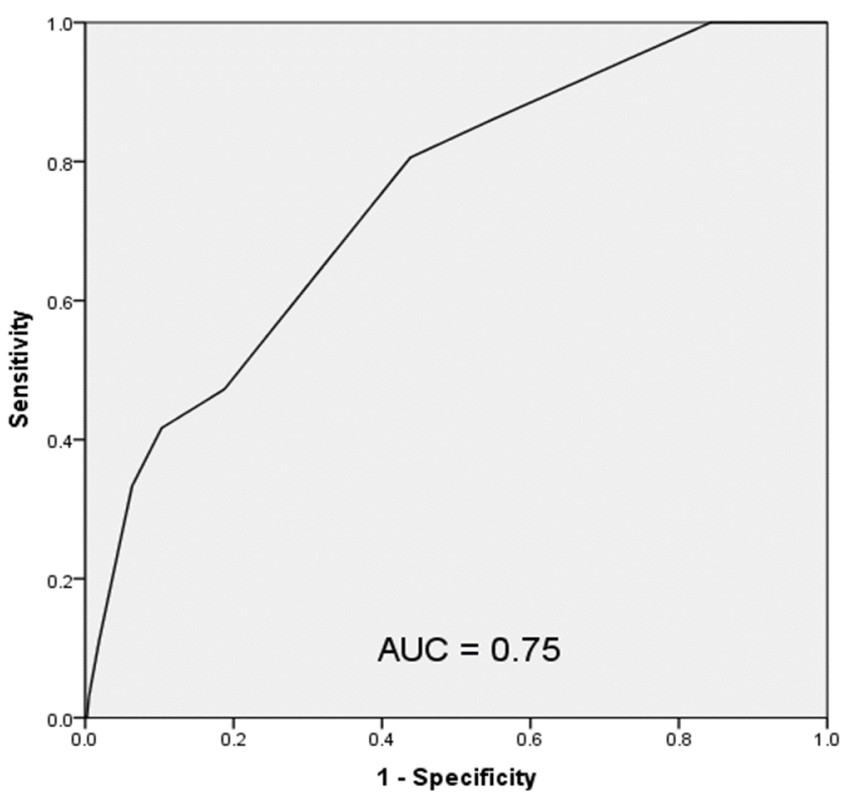

**Figure 2** **Receiver operating characteristic curve of the HOSPITAL score in the study population.**

(*Van Walraven et al., 2010*) and data from a nationwide Medicare cohort (C statistics of 0.58 vs. 0.680) investigating all cause hospital readmissions (*Garrison et al., 2016*).

The difference observed between our study and the validation study in the discrimination of the LACE index of predicting readmission can be explained by the differences between the study populations. The original validation study as compared to our data had much lower Charlson comorbidity index scores (mean of 0.5 vs. 4.47), lower number of emergency visits (mean of 0.4 vs. 2.36) and large sample size (1,000,000 patients vs. 432) (*Van Walraven et al., 2010*). It is also important to mention that 44.9% of patients included in the original validation study primary cohort were admitted to a medical service (*Van Walraven et al., 2010*), as compared to our study which has 100% patients admitted to medical service.

The nationwide Medicare cohort study differs substantially from this study by including all hospital admissions (emergent, elective, medical, and surgical vs. emergent and medical in this study) and is limited to Medicare beneficiaries (*Garrison et al., 2016*). This study includes all adult patients, regardless of insurance status.

The LACE index was derived from a small sample of patients (approximately 2,500) done at the original validation study (*Van Walraven et al., 2010*). The LACE index in our study population had an overall good performance with a brier score of 0.082 and a Hosmer–Lemeshow goodness of fit test showing a $\chi^2$ value of 4.97 with a *p* value of 0.66.

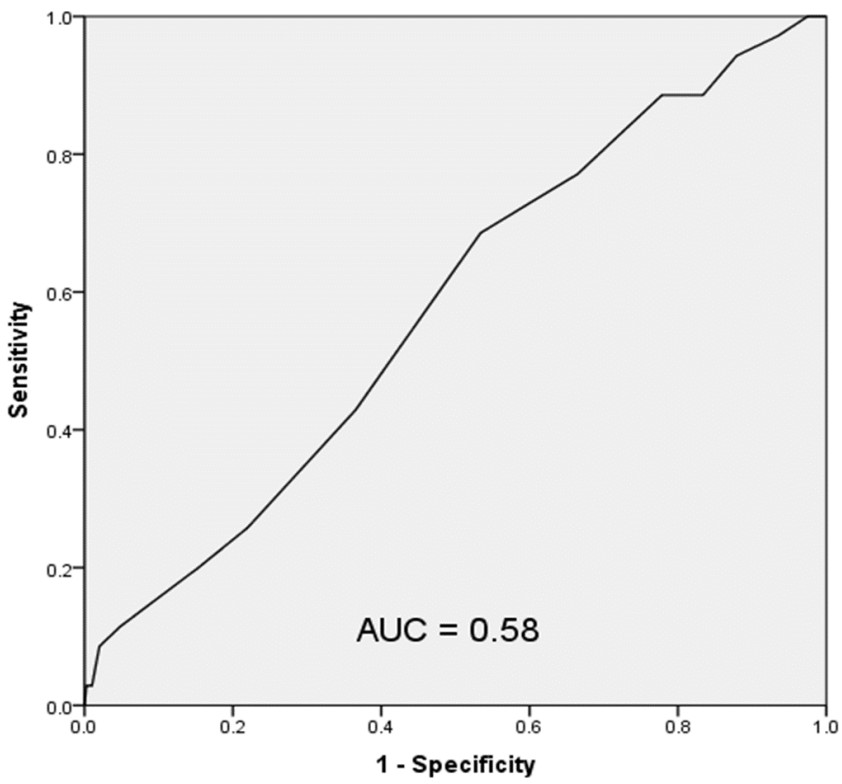

**Figure 3   Receiver operating characteristic curve of the LACE index in the study population.**

The goodness of fit test was superior in the original validation study of LACE index with Hosmer-Lemeshow statistic of 14.1, *P* value of 0.59 (*Van Walraven et al., 2010*).

The study population differs from the international validation study of the HOSPITAL score in several important ways. The study hospital does not have a distinct oncology admitting service, all of the admissions during this timeframe were classified as urgent or emergent, and discharge day laboratory tests for hemoglobin (11% vs. 94%) and sodium (53% vs. 97%) were less frequently performed (*Donzé et al., 2016*). The derivation and international validation studies accepted the last laboratory tests for hemoglobin and sodium as the values at the time of discharge, this study only accepted results for tests performed on the day of discharge for these predictor variables (*Donzé et al., 2013*; *Donzé et al., 2016*). These factors are due to the local practice environment at the study site and are likely to have resulted in lower HOSPITAL scores for some discharges. This would lead to a reduced accuracy of the HOSPITAL score to predict readmission in this environment. Despite these differences, the HOSPITAL score performs well at this moderate-sized university-affiliated hospital.

The poor performance of the LACE index observed in our study was also shown on one study done in UK (*Cotter et al., 2012*). Similar to our study, the cohort studied had longer hospital stays and higher comorbidities as compared to the original validation study (*Van Walraven et al., 2010*). The authors concluded that more details are needed and should be

added to the LACE index to improve its performance in predicting readmission (*Cotter et al., 2012*). Multiple other studies validated the moderate discrimination and good fit of the LACE index of predicting readmission (*Garrison et al., 2016*; *Spiva et al., 2016*; *Van Walraven et al., 2010*).

The focus of all cause (avoidable and unavoidable) hospital admissions is a different endpoint than the potentially avoidable readmissions investigated in the derivation and validation studies for the HOSPITAL score (*Donzé et al., 2013*; *Donzé et al., 2016*). The endpoint of all cause readmissions is highly relevant because it is a significant marker of hospital quality under the Medicare VBP program for hospital reimbursement. Under this program, hospitals with high readmission rates can face financial penalties. To improve performance for this key healthcare quality measure, hospitals and health systems could use the HOSPITAL score to identify patients that may benefit from interventions directed at reducing hospital readmission. The HOSPITAL score is suitable for adaptation into an automated clinical decision support tool within an electronic health record system to identify patients at increased risk of hospital readmission. Finally, the readmission process is complex and multifactorial, this notion can be derived from the evidence that no single intervention was shown to be adequate alone in preventing readmission, as show on a recent meta-analysis (*Leppin et al., 2014*).

This study has several important limitations. This study and the international validation study for the HOSPITAL score share an important shortfall by only identifying readmissions within 30 days at the same hospital (*Donzé et al., 2016*). This limitation was overcome in the LACE index original validation study by contacting the patient 30 days after discharge of the hospital to determine the readmission status (*Van Walraven et al., 2010*). Furthermore, this study is retrospective, single center, focused on medical patients, small sample size, and shaped by local practice patterns (no oncology admitting service, few elective admissions, infrequent laboratory testing on the day of discharge). These limitations may reduce the generalizability of these results.

This study shows that the HOSPITAL score is useable in moderate-sized community-based hospitals to identify patients at high risk of readmission. Whereas the LACE index has poor performance in identifying the readmission risk in complex medical patients with increased length of stay. Identifying these patients for interventions targeted at reducing hospital readmissions may result in improved patient care outcomes and healthcare quality.

## CONCLUSIONS

The internationally validated HOSPITAL score may be a useful tool in moderate-sized community hospitals to identify patients at high risk of hospital readmission within 30 days. This easy to use scoring system using readily available data can identify patients at high risk for hospital readmission. The LACE index was shown to be not adequately validated in a moderate-sized hospital to identify the risk of readmission of complex medical patients. Further identifiers are required in addition to the LACE index score is required to improve performance in such population. These patients could then be targeted with interventional strategies designed to reduce the rate of hospital readmission.

Further research is needed to determine if the HOSPITAL score and LACE index score are useful as a readmission risk prediction tool in other patient populations.

### Funding
The authors received no funding for this work.

### Competing Interests
The authors declare there are no competing interests.

### Author Contributions
- Robert Robinson and Tamer Hudali conceived and designed the experiments, performed the experiments, analyzed the data, contributed reagents/materials/analysis tools, wrote the paper, prepared figures and/or tables, reviewed drafts of the paper.

### Human Ethics
The following information was supplied relating to ethical approvals (i.e., approving body and any reference numbers):

Institutional review board review for this study was obtained from the Springfield Committee for Research Involving Human Subjects. This study was determined not to meet the criteria for research involving human subjects according to 45 CFR 46.101 and 45 CFR 46.102.

### Data Availability
The raw data has been supplied as Data S1.

### Supplemental Information
Supplemental information for this article can be found online at http://dx.doi.org/10.7717/peerj.3137#supplemental-information.

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
