# Peer review of "The HOSPITAL score and LACE index as predictors of 30 day readmission in a retrospective study at a university-affiliated community hospital"

_PeerJ, doi:10.7717/peerj.3137_

## Round 0.1 · original submission · Major Revisions

Dear authors,

Thank you very much for the opportunity to handle this paper. After reading it with the reports of the reviewers, I think it has scientific merit to be published in PeerJ, once some issues are solved by you. Therefore, my decision is MAJOR REVISION.

With respect and warm regards,
Dr Palazón-Bru (academic editor for PeerJ)

Reviewer 1 ·

Basic reporting

The article is very well written, clear, well-structured and using professional English language.
The study background and context is thoroughly described in the introduction, and adequately referenced. I would suggest on line 29 of the introduction to replace “The underlying causes...” by the “The underlying risk factors…”, since most of the factors mentioned are not a direct cause for readmission.
The author mention a previous that compared the HOSPITAL with the LACE index (Garrison 2016), but failed to mention 2 other studies that compared these 2 scores: a) one in Denmark with around 20,000 medical patients. the HOSPITAL score significantly overperformed the LACE index (Cooksley QJM 2015); b) one prospective study in Switzerland, where the C-statistic was 0.70 for the HOSPITAL score and 0.56 for the LACE score (Aubert, Swiss Medical Weekly 2016). Please add these 2 other studies, either in the introduction, or in the discussion. It could be also highlighted that in the Garrison study, the patients were from the ambulant setting, and readmitted to any department, as opposed to only medical patients.

Please add in Table 3 the number and percentage of patients identified with a diagnosis of cancer using the ICD codes (as surrogate for the “discharge from oncology”), along with the p value.
As optional minor modification, the author might try to merge both figure 2 and 3, in order for the reader to better appreciate the comparison between the ROC curve of the HOSPITAL and LACE scores.
Line 190: the C-statistic was 0.75 in this study, not 0.77, please correct.

The figures and tables are relevant and of good quality.

Lines 233-235: some other studies

Experimental design

The study is original, and bring valuable information for the practice.

Validity of the findings

The data collected are appropriate for the study question.
Discussion is well-written and include the required information.

Additional comments

In this retrospective cohort study, the authors aimed to validate a prediction model (the HOSPITAL score) to identify patients at high risk for 30-day readmission. The populations studied is medical patients discharged from a moderate sized university affiliated hospital in the midwestern United States. In this population, the HOSPITAL score showed good performance. The article is well-written and methodology sounds.

Reviewer 2 ·

Basic reporting

The authors explored the performance of HOSPITAL score and LACE index to predict all cause 30-day readmissions using 432 discharges from a university affiliated community hospital.

Main concerns:
(1) This paper conducted a similar comparison study as Garrison, 2016. But the conclusions are very different. In Garrison , 2016, their C-statistics were around 0.68 for both HOSPITAL score and LACE index; in this study, they were 0.75 and 0.58 and the difference was very significant. It would be important to understand the possible explanations for such a big difference.
(2) The authors in this paper showed that their results is comparable to the results from Donze et al, 2016. But I don't think it is a fair comparison considering that the readmissions here are all-cause and in Donze's are potentially avoidable readmissions.

Minor concerns:
(1) In Line 190, the authors mentioned the C statistics of 0.77 vs. 0.72. However, the C-stat seems 0.75 in this paper and 0.71 in Donze, 2016.
(2) Figure 2 and Figure 3 can be in the same plot to be concise and to compare explicitly.

Experimental design

Major concerns:
(1) It is not mentioned in this paper that how many multiple readmissions from the same patient are considered, which I think is very important, especially considering here the smaller sample size and higher hospital admissions for the same patient. Specifically, I would suggest the authors to explicitly show how many distinct patients there are in the in 432 adult discharges, and how many distinct patients there are in the 35 readmissions. The authors need to show what to do to make the results valid if multiple readmission is in deed a concern in the data.

Validity of the findings

No comment.

Additional comments

There are several paragraphs with only single sentence in the section of Materials and Methods and the section of Statistical Analysis. I would suggest the authors to combine some of them to make the paper neat and nice.

---

## Round 0.2 · accepted · Accept

Dear authors,

After reading your revised version of the text, I think it has high standards to be published in PeerJ. Therefore, my decision is TO ACCEPT.

Congratulations!

With respect and warm regards,
Dr Palazón-Bru (academic editor for PeerJ)